# In employees' favour or not?—The impact of virtual office platform on the work-life balances

**Dharshana Rathnaweera[1], Ruwan Jayathilaka[2]\***

1 SLIIT Business School, Sri Lanka Institute of Information Technology, New Kandy Road, Malabe, Sri Lanka, 2 Head—Department of Information Management, SLIIT Business School, Sri Lanka Institute of Information Technology, New Kandy Road, Malabe, Sri Lanka

\* ruwan.j@sliit.lk

**Data Availability Statement:** Data cannot be shared publicly because of data protection. Data are available from the Sri Lanka Institute of Information Technology, SLIIT Business School,

## Abstract

Social mobility and physical restrictions imposed to contain the spread of the COVID-19 pandemic have posed a severe challenge to operate under conventional nine to five work practices in a physical office setting. As a coping strategy for the survival of business, economy, and livelihoods, certain organisations were compelled to transform to virtual office platforms. This was a sudden transformation of work practices and consequently, impacting with mixed outcomes on lifestyles of workers. Given that very limited studies have shed light into the context, this study contributes immensely to fill research gap. The main objective of this study is to identify the impact of the virtual office platform on work-life balance in the Sri Lankan context. The methodology adopted for this study is quantitative. An online questionnaire to collect data was primarily distributed to employees in the virtual platform. Analysis of this study is based on three regression models and results ascertain that both working and non-working environments have highly significant impact on the work-life balance, although non-working environment has a bigger influence on work balance (Gender and no of children). Findings are useful and unique, enabling both employers and employees to adopt a focused approach to maximize the potential of virtual platforms to enhance employee well-being so that mutual benefits can be materialized.

## Introduction

Crosbie and Moore [1] state that homeworking or working from home can improve the work-life balance. Also employees who are working from residence are in a better situation, as those who desire to work from home have higher levels of job satisfaction, and have a better work–life balance under a rigorous contractual arrangement than with a soft law commitment [2]. Employees are the most valuable resource that can impact organization's success and outstanding performance. Hence, it is important to maximize the potential and the quality of human resources, addressing and resolving employee-related issues in a proactive manner. Work-life balance is crucial for both the organization and its employees from a strategic standpoint since people play multiple roles as every person is a part of their families and for organizations they

Sri Lanka (contact via Prof. Samantha Thelijjagoda, Dean, SLIIT Business School, Sri Lanka Institute of Information Technology, New Kandy Road, Malabe, Sri Lanka. Email: samantha.t@sliit.lk) for researchers who meet the criteria for access to confidential data.

**Funding:** The authors received no specific funding for this work.

**Competing interests:** The authors have declared that no competing interests exist.

work. Individuals desire to change their work patterns so that employees themselves can spend adequate time focusing on personal tasks for work-life balance.

The contemporary world is known as a Volatility, Uncertainty, Complexity, Ambiguity (VUCA) one, where no one can envisage the next wave of global changes that can unexpectedly hit the world. The coronavirus (COVID-19) outbreak is the most recent change, the global economic shock that hit the hardest as the pandemic turmoiled operations in all countries. At the time of writing this research paper, COVID-19 has negatively affected the livelihoods and daily lifestyles of the majority of people worldwide. Almost all businesses had to close operations permanently or temporarily. Business entities were compelled to downsize to run with fewer employees and facilities as a collapse of business markets shrunk the revenue-generating avenues. This situation further tightened challenges for the continuity and survival of businesses. However, certain larger organizations or categories of business sectors that have had sufficient profits and resources to face this unprecedented challenge, managed to continue their business operations in a virtual platform. To handle this situation effectively, such organizations transformed a considerable portion of their critical operations and services, resulting in a shift of work practices–from a physical office setting into a virtual platform. To keep businesses afloat and adapt to this situation triggered by uncertainty, organizations introduced electronic devices by providing all kinds of necessary tools to carry out remote working, referred to as 'work from home (WFH). It can be highlighted, during this time, novel business opportunities, product, service markets, etc., also emerged and evolved via e-commerce, thus further convincing the online virtual platforms is desirable. As with the rest of the world, Sri Lanka too faced economic setbacks due to the pandemic and had to switch to WFH for survival of the economy.

Telework and virtual office are defined as "simply the use of telecommunications-related technology to conduct work" [3]. This WFH concept was limited mostly to Information Technology (IT) firms and those that operate on IT-based processes and procedures. As such, along with the COVID-19 outbreak, organizations of this kind widely applied WFH in Sri Lanka as well, which impacted the lifestyle of employees both negatively and positively, depending on their atmosphere at home [4, 5]. Even certain employees of generation "X" [6] were affected as their jobs were redundant or obsolete, since these employees could not adapt to the new technology; some of them were also reaching their retirement age. In the virtual office [7], employees are provided with portable devices to accomplish their job and are often granted the flexibility and the authority to work from wherever feasible as they prefer, to accomplish business objectives. Minimal research has been conducted with workers in the virtual office setting. Unlike in a physical office setting, trust and a high level of communication between employees of all levels are critical for effective functioning of a virtual office. Communication plays an important role during a pandemic of this magnitude. Communication needs to be bottom-up and both ways as the management should be aware of what to do and what not to do, before implementing or introducing new ways and means of working [8]. If communication fails, it will negatively affect the motivational and productivity level of employees, which will, in turn, affect organizational performance, strongly and adversely. The organization should obtain the feedback of employees for a successful transformation in business operations for the productive functioning of the home-based work platform [9].

Adoption of this new change is a massive issue, a giant leap that needs much consideration. Not all employees are rich in knowledge on technological advancements, electronic equipment, new systems, etc., which are the factors highly engaged in the virtual office platforms. Similar studies focus on two of the several possible performance-improving strategies explored in different fields (such as business, management, psychology, and family sciences) the reduction of work-family conflict (WFC) and the increased use of telework [10]. On the one hand,

those who have facilities required for a virtual platform and organizations which can provide such facilities could handle this situation or similar, without experiencing it as a major issue. On the other hand, with the loss of jobs, shutting closure of organizations (employers), and loss of earnings, small businesses and old school employees are severely affected beyond what one could imagine.

Many advantages and disadvantages are associated with the virtual office platform on the work-life balance. Women are the most affected by the virtual office platform, especially those with kids [11, 12]. Such women have to meticulously manage their time to share work and family responsibilities, to maintain a proper balance of work and personal life. Some evidence indicates that a large number of organizations increase the benefits of employees concerning family benefits, to improve the productivity level of employees with some motivational factors. However, in general, studies justify that virtual office platform has both negative and positive impact on work-life balance [13, 14] certain factors have critically affected some families due to the new method of working virtually. Yet, there is a gap between some findings in some research. The present study mainly considers the impact of the virtual office platform on the work-life balance in the Sri Lankan context.

## Problem statement

Along with the COVID-19 scenario and social distancing and mobility restrictions, working hours and workplaces were forced to change, specially the conventional eight to five work practices was no longer feasible. Companies are adopting flexible work schedules as they have allowed employees to work remotely from home during lockdown times. In particular, the companies had to incorporate flexible work schedules to preserve the mental well-being of employees as well as mental and physical breaks, workouts, indulge in other non-work activities, minimize anxiety and improve the degree of efficiency. Therefore, it is necessary to identify effects of the main factors that affect family life during the WFH period. Limited research publications are available on this area of research in the Sri Lankan context. The authors conduct the present study by addressing the above-mentioned scenario as a timely problem. Therefore, identify the nature of this problem under the research topic on the impact of virtual office platforms on the work-life balance is of significance.

## Empirical hypothesis

There are two hypotheses in the proposed model. The hypothesize correlations on the proposed model will be justified in the following manner, as indicated in the literature survey. Hypotheses are as follows.

Hypothesis 1: There is a positive effect of working environment on work-life balance.

Hypothesis 2: There is a positive effect of a non-working environment on work-life balance.

## Rationale of the study

The findings of this publication allow a pragmatic and convenient approach to identify the impact and the magnitude of work-life in a virtual office platform. This publication will carry unique findings in a Sri Lankan work environment, which will assist the employees as well as the employers.

First, the present study enables employees to identify the areas that need to be considered when moving into a virtual office platform. The employers also will be better aware of when

their employees need to be moved to different work platforms and what kind of minimum facilities or requirements be provided to the employees for successful transformation—from a physical work environment to one that is virtual.

Second, particularly in light of the COVID-19 pandemic, it is critical to investigate the effect of the virtual office network on employees. As noted previously, this study supports both parties, i.e. employee and employer, to identify and adapt to a new working environment by considering the critical factors indicated above.

Third, there has been very limited prior research project related to the Sri Lankan context that highlights on future implications and policymaking related to the COVID-19 pandemic situation. Finally, the results would be useful to various industries and policymakers in an endeavor to recover from economic setbacks and improve quality of life, following the global COVID-19 pandemic outbreak in early 2020.

## Objective

The main objective of this study is to identify the impact of the virtual office platform on work-life balance in the Sri Lankan context. As a result, this study differs from previous researches and adds to the existing literature in above four aspects. In the beginning, there were few studies in this field in Sri Lanka.

## Literature review

This study centered the literature review based on the initially defined 55 articles through a comprehensive and detailed literature search. Several reputed journal databases such as Science Direct, Taylor & Francis Online, Emerald insight, Taylor and Francis online, Research gate, IEEE Xplore, Social Science Research Network, Google Scholar, and Wiley online library, etc., were referred. The search term used were (Work-life balance or virtual office or virtual platform or WFH or work life or telework).

For a better understanding, these 55 articles searched were segregated into several sections. The flow diagram in Fig 1 depicts the series of studies found, retained, and dismissed through each phase of the literature review. Accordingly, this literature review was classified into three main sections. Hence, the above search articles were also classified into three (03) main aspects as work-life balance (WLB), working environment (WE), and non-working environment (NWE), as per the variables identified by the conceptual framework.

## Work-life balance

Hobson [15] specified that connection among parents as well as a career is referred to as work-life synergy. Balven, Fenters [16] found out organizational assistance for personal life components is referred to as work-life balance. Under work-life balance to measure family satisfaction and identify the relationship, lifestyle, and family personal interests, Kopelman, Greenhaus [17] focused on the topic of validation of structures, purposely forgoing investigation into various intriguing and substantive problems. Life satisfaction is also another factor to determine the work-life balance and a non-supportive working environment may lead to a high level of work-family conflict. Moreover, Greenhaus, Bedeian [18] observed a negative relationship between job performance and marital adjustment including work-life balance among females. A recent survey focused primarily in Europe shows that work-life balance prospects are unequal, as well as gender disparities: males seem to be quite happy at work balance than females [19]. Hill, Ferris [8] illustrate that the link among telework and work performance is ambiguous. Both virtual office and home office workers believed that telework had

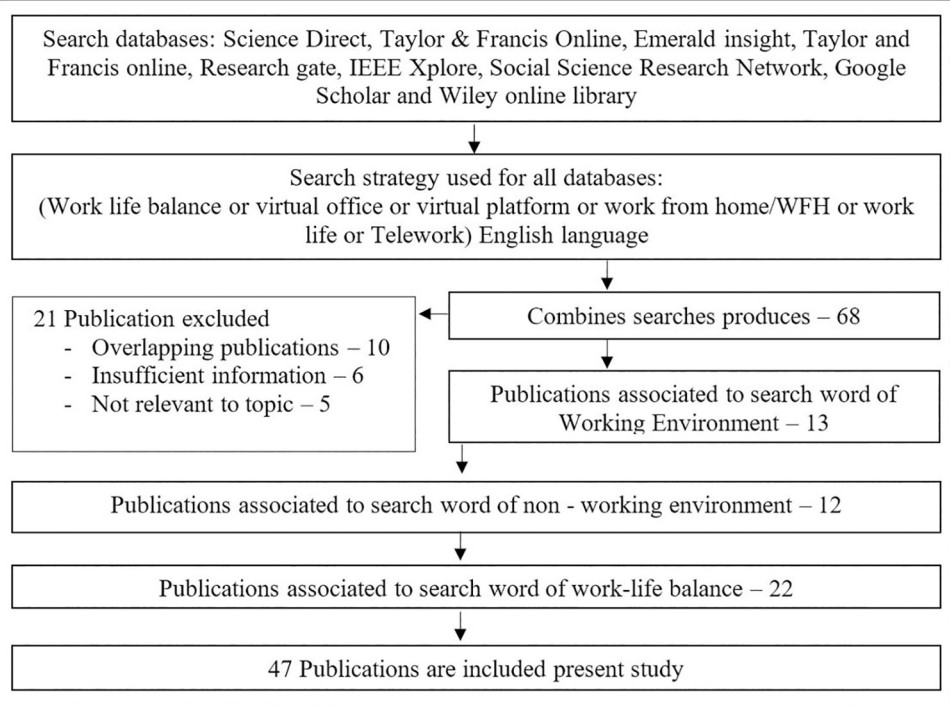

**Fig 1. Literature search flow diagram.** Source: Based on authors' observations.

increased their overall performance, whereas the home office workers seem to be the most optimistic.

Hilbrecht, Shaw [20] conducted a study in the Canadian region and investigated whether virtual work is especially advantageous for females. Data indicated that telework facilitates intensive motherhood practices, further limiting the recreational opportunities for women. It could be the involvement of men rather than women in telework that could theoretically be a significant factor in the restoration of gender relations in the oncoming periods. Having a positive social undermining is another factor associated with the occurrence of a positive work-life balance. In a study based in Sri Lanka, Malalasekara [21] examined the relationship between work-life balance and employees' workplace performance. Findings discovered that employee-related aspects of family roles significantly affect the success of both private and government sector workers. Childcare concerns, dependent care, and workers with healthy time with the family indicated a favorable and important linear relationship with performance metrics such as consumer satisfaction, target expectations, and employee satisfaction.

## Working environment

It is a widely held belief that the atmosphere of the workplace or the work environment has a variety of effects on the workforce's general mental and physical health, particularly in light of the current pandemic situation on both the work–home fronts [22]. Frone, Russell [23] found out the factors associated with the working environment were negatively related to both measures of work-non-work conflict. Moreover, results indicated that the traditional working environment is highly influenced by non—work factors, while the quality of life is primarily influenced by work factors.

Family conflict is one of the indirect key factors that fall under the purview of the working environment. Family conflicts can be measured and identified through relationship bonds,

lifestyle, and personal interests. By studying and analyzing these measurements, it is easy to identify whether there are conflicts or not. For analytical purposes, a robust work-family interface model was built and validated by Frone, Russell [23]; further, this model expanded previous research by specifically distinguishing between work that interferes with family and family that interferes with work. In terms of putative causal factors, the model was conceptualized and the correlational evidence does not enable causal intervention to be drawn about the separate hypothesized relationship. To look into work-non-work conflict and perceived quality of life study, Vong and Tang [24] proved that when one's professional experience compromises family life, work-family conflict arises. Various work-related variables have most likely contributed to the creation of Conflict of work-family.

Furthermore, homeworking space is also another factor that is critical to consider while working on a virtual platform. Hamblin [25] asserted that virtual work options for such workers might need to be consciously considered on the employer's human resources policy in addition, the study demonstrates workers' expectations on what constitutes flexibility can vary from the practices that a company may choose to adopt. In another study, Habib and Cornford [26] proved that home-based telework needs to be investigated periodically. Looking at teleworking from an inclusive view is more than just a functional aspect of an incentive for the staff at the individual level; however, teleworking is a significant transformation in lifestyle that can impact the entire family unit. Frone, Yardley [27] carried out a study to develop and test an integrative model of the Work-Family Interface and findings indicated that family-to-work conflict was negatively related to work performance, although the work-to-family conflict was negatively linked to family performance. The analysis extends beyond those findings of previous studies by demonstrating three proximal predictors in each form of conflict (i.e., distress, overload, and time commitment). Our model thus clarifies the meditational pathways that connect more other distal predictors to conflict between work and family.

Hill, miller [7] compared virtual office teleworkers and an analogous group of conventional office employees statistically. For this study, sample group forces focus mainly on IBM (International Business Machines Corporation) employees. The issue was that IBM employees were more compliant with the telework environment than employees of other businesses. The methodology of the questionnaire alone does not sufficiently capture appropriate dimensions of the virtual workplace. Interviews and findings as well as questionnaires, may also provide potential information and knowledge that could useful for further investigations. That said, the study will be more comprehensive and generate reliable findings. The discussion of results is structured around findings that generalize both qualitative and quantitative approaches (productivity, flexibility, and balance of work-life) and then those (morale, teamwork, and hours worked) did not generalize any approaches. The purpose of this study to discover the proof of how the employment relationship is shaped by teleworking and the consequences this may have for those line managers accountable for a home-based workplace. Harris [9] found that before adopting home-based working, managers should recalibrate their views of the boundaries between home and work to facilitate productive employee interactions within a modern model of "home-work" relationships. A known drawback of this research is the study explores the employee experience of home-based teleworking since most adaptation issues are likely to encounter during its initial year of service.

Research conducted in Iceland proves that participants are happy to give up their current work schedule for a 9 to 5 task (regular conventional working hours) [28]. This study conducted in Iceland included selecting 10 semi-structured, in-depth surveys with academics. In a more extensive mixed-method analysis on the work-family balance, work arrangements, and the well-being of academics, for the interviews. Ten male and 10 female respondents were chosen based on being a parent, holding a permanent role, ranking as a senior lecturer, associate

professor, or a professor, and working in one of the three largest university areas of expertise; if the author chooses more of the population as a sample, it must be more precise.

Moreover, studies have already proved that space used for teleworking at home has an impact on conflicts between work and family. Besides, having other people at home while teleworking harms family time for individuals; those who have an exclusive workspace encounter less family interference with work few hours per week [14]. Moreover, research data were primarily based on teleworkers only.

## Non-working environment

Knowledge workers can now operate from everywhere and in multiple areas unlike in a one permanent area, as a result of advanced communication technology and mobile devices (ubiquitous working). To identify if the environment affects decision-making and focus of employee performance, the researchers varied the environment (i.e., a virtual office as a normal work atmosphere vs. a virtual garden as a non-work environment) and time pressure (i.e., introducing time pressure vs. no time pressure) [29].

However, numerous pieces of evidence indicate that there is an impact on the gender aspect and the number of children mainly affects the virtual performances in the non-working environment. Yap and Tng [30] discovered that most female computer professionals in Singapore prefer to telework just 1 to 3 days a week. For instance, this fact is especially valid in times of need, when such women have children. Also, Hamblin [25] indicated that motivation towards virtual work could be noticed among females due to the after-school and school holiday childcare.

Grzywacz and Carlson [31] considered work-family synergy as an achievement of role-related aspirations agreed and shared by a person with his or her role-related colleagues mostly in the work-family interface. Wong, Cheung [13] showed that personal family well-being enhances WFH effectiveness, although environmental resource barriers reduce it. Therefore, after the pandemic, employees who have experienced stronger WFH performance indicate their maximum preference for WFH.

Lim and Teo [32] discovered that males and females do not vary in their teleworking approaches. These findings postulated that women tend to have a more favorable teleworking attitude than men. Preference for working at home was linked to the gender factor, with a higher percentage of women preferring the teleworking option as opposed to males. Findings indicated that married workers would have a more beneficial approach to teleworking compared to staff who are unmarried.

Furthermore, Sullivan and Lewis [33] asserted that how the female life becomes easier comparing the traditional work schedule, mainly due to that virtual work schedule and flexible work schedule platform. With the help of telework, both employees and employers can integrate with the demands of their domestic as per their requirements, mainly their household chores and any kind of family matter [34]. Article findings also point out limitations. Some biases in the respondent group were evident, and the response of some participants can be accepted. It must be noted that all respondents were public sector employees. Within such organizations, employment arrangements are more likely to be regulated, and work-family patterns are more likely to have been operating for a specific period, thus producing an organizational culture that is more 'family-friendly'. The respondents, like most teleworkers, were engaged in some type of white-collar work. Numerous shreds of evidence indicate that telework will contribute to the balance of life, mainly on Canadian working women, in a positive way. These female workers can relax more in their working environment and also reduce their stress levels and pressure [35]. Besides, the scope of the study is restricted to the telework experience of women, thus considering the impact of men teleworkers overlooked.

Nevertheless, to study work-life balance–an issue less studied among Austrian female expatriates, Fischlmayr and Kollinger [36] conducted a study. where they came up with a work environment that has a more substantial impact on the family, with the family life, and on work. These researchers also found out that there is a relationship between female workers and work-life balance.

Various studies were conducted to assess the gender effect on virtual platforms. Calvo-Salguero, Salinas [37] are of the view that there are no gender disparities in family interfering with job duties since the time spent on the family affects the degree to which women encounter the conflict. Men spend more time on work than women and gender differences in work-family responsibilities mainly occur depending on the number of time women spend on their family, in comparison to men. Home-based working is anticipated to expand and diversify in nature, hence as a result, this will remain to pique the interest of academics and practitioners alike [38].

Moreover, Solís [39] investigated that level of responsibilities that lies with each individual (Male and female) outside the working environment mainly affect work-family relations. He further stated that employers who perform in virtual office platforms with a high level of responsibilities have a high level of work interference with the family. A study conducted based on female Spanish teleworkers, Gálvez, Tirado [40] concluded that before they commence on the virtual office female Spanish teleworkers have to negotiate with these elements (Family attitude to the work, attitude to the work commitment, and material arrangement).

Furthermore, in many cases, gender and children have a direct link with virtual office performances and family life. Zhang, Moeckel [41] claimed that teleworking includes more married men without children than married women without children. Sex and marital status are less likely to telework individuals with children than their peers without kids. It is noteworthy that married women with kids aged 0–5 years are more likely to telework than married women without children.

This segment provided a summary of the effect of the virtual office on family life. By highlighting the key features of a large number of selected studies, this study also examined the accumulated sizeable empirical literature on the determinants of the effect of the virtual office and those related to work-life balance. The majority of the literature reviewed in this part has revealed several efforts to determine the effects of family life. As can be seen from the literature review, almost all the studies have concentrated on various aspects of family life and virtual offices. However, none of these specifically examine how virtual offices affect work-life balance. According to some reports, negative effects were reported primarily due to virtual workplaces, but these were unrelated to family life. There has been no systematic study or connection to quantify the effect of a virtual office on work-life balance in studies that have considered the impacts of the virtual office and well-being family-life separately. The present study is unique since it considers in a combined scenario, the characteristics and the impact of virtual office and work-life balance. As a result, it is vital for every company, worker and, a family member who struggles with the effect of a virtual workplace on work-life balance to address this policy problem.

Nevertheless, there is a small body of literature on the effects of virtual offices and work-life and no local contribution. Based on the lack of a structured method and research in describing the impact of a virtual office on work-life balance, the key concern of this study is to discover new findings to fill the above-mentioned research gaps. Thus, contributing to this void is accomplished by systematically analyzing the inherent association effect between the virtual office and family-life balance using various analytical methods.

## Methodology

This study was reviewed and approved by SLIIT Business School and the SLIIT ethical review board. Data were collected using online forms and conducted individual interviews online.

The questionnaire comprised of two sections. The first section intended to gather demographic data, while the second section intended to obtain additional data required for the measurement factors. The purpose of the questionnaire was to operationalize variables, including all variables considered in the conceptual study. A minimum of one question and a maximum of four questions were assigned for each variable. The questions were developed with a five-point Likert scale to measure each model variable, on an ascending scale from 1 to 5, depicting (1) strongly disagree to (5) strongly agree and the questionnaire is presented in S1 Appendix.

Each individual in this study confirmed their verbal consent prior to the formal online interview. The researcher proposed a quantitative methodology to conduct this study. The population of the study explained the total number of elements that focus on the research study. The researcher applied a random sampling method, mainly employees who are working on a virtual platform to collect the required data and other information. The sample size was selected using online sample size calculator *goodcalculators*. The margin of accepted error was taken as 5% and confidence level of 95%. The results indicated that sample size of at-least 270 would be required. This sampling technique aided in the collection of data that was free of bias. Pilot survey of 30 respondents were conducted to determine whether the questions were clear, understandable, and logical order. The researcher distributed questionnaires by using Google forms; conducted individual interviews online, and carried out direct observations, and gathered more reliable information. The survey was distributed to participants by sharing the online link of the survey through prominent social media networks and email. The sample of this study consisted 270 employees from organizations in various sectors (Lecturers, students, upper management, middle management, and line management) located in the Colombo and Kandy districts (Private companies, banks, educational institutes, IT companies, and government institutes) and data file is presented in S2 Appendix. Employees of these districts were keen to continue operating in the virtual office platform along with the Covid'19 situation, hence, were identified as appropriate to carry out the study [42].

Analysis of responses indicated the mean value of variables collected while using the three models. Multiple linear regressions were used for the first (1) model considering dummy variables, multiple regression analysis for variables (2), and finally, a forward stepwise regression technique was used for model (3) to develop the relationship among the variables. The following is an explanation of the process of regression formulas.

## Analytical tool

Unless the independent variable is known, regression analysis helps predict the dependent variable. This regression tool assists to determine variables which have the most significant effect on employee results. The regression analysis output is segregated into three models. Since there are two independent variables in this analysis, the researcher used multiple regressions and the forward stepwise regression technique.

$$WLB_i = \beta_0 + \beta_1 WE_i + \beta_2 NWE_i + \beta_3 DM_i + \varepsilon \qquad (1)$$

where for $i$ = n observations:

In this model, as $WLB_i$ depicts work-life balance and $WE_i$ and $NWE_i$ depict a working environment and a non-working environment, respectively, in addition to $DM_i$ which indicates dummy variables. Based on exploratory research, variables to include in the empirical specification of this study are determined.

The significant variables are chosen using the forward stepwise regression technique. With a p-value of 0.05, new variables are selected, and previously selected variables are removed with a p-value of 0.10. The model's goodness-of-fit is assessed using [41] a developed overall

goodness-of-fit metric; the model with the highest goodness-of-fit value is chosen for this evaluation.

## Results

Descriptive statistics (measures of central tendency, measures of dispersion, and frequencies of individual levels) were used to accomplish the research goal, i.e., to determine the effect of the virtual office platform on work-life balance. Gender reflected various levels of responsibility and obligations for employees' families and workplaces. As a result, the researcher gathered knowledge to comprehend actions of participants (total 270 nos.) in the study. Besides, Table 1 shows a general description of the demographics of the respondents used for regression analysis.

### Reliability results

Information that is reliable and accurate enables researchers to arrive at a meaningful analysis. Further, reliable information has a significant impact on the study's outcome. As such, the

**Table 1. Definitions of variables with mean values.**

| Variables | Description | % (Mean values if numerical) |
|---|---|---|
| WLB (dependent) | The average value of family satisfaction, marital satisfaction, life satisfaction, social undermining, personal growth, depression and distress, daily alcohol consumption, and physical health | 3.2704 |
| WE (independent) | The average value of home working space, Number of the person in the house, Working days, Responsibility and, Time duration | 3.5726 |
| NWE (independent) | The average value of gender and number of children | 3.3637 |
| Age (20–25) | Dummy variable = 1 consider as age (20–25) and 0 for otherwise | 12.6% |
| Age (26–30) | Dummy variable = 1 consider as age (26–30) and 0 for otherwise | 39.3% |
| Age (41–50) | Dummy variable = 1 consider as age (41–50) and 0 for otherwise | 10.4% |
| Gender (Female) | Dummy variable = 1 consider as gender (Female) and 0 for otherwise | 51.9% |
| Education (GCE A/L) | Dummy variable = 1 consider as education (GCE A/L) and 0 for otherwise | 3.3% |
| Education (Diploma) | Dummy variable = 1 consider as education (Diploma) and 0 for otherwise | 14.1% |
| Education (Degree) | Dummy variable = 1 consider as education (Degree) and 0 for otherwise | 51.5% |
| Education (Post Graduate) | Dummy variable = 1 consider as education (Post Graduate) and 0 for otherwise | 3.7% |
| Monthly Income (Below 50,000 Sri Lankan Rupees (LKR)) | Dummy variable = 1 consider as monthly income (Below 50,000 LKR) and 0 for otherwise | 8.5% |
| Monthly Income (100,000–150,000 LKR) | Dummy variable = 1 consider as monthly income (100,000–150,000 LKR) and 0 for otherwise | 24.8% |
| Monthly Income (Above 150,000 LKR) | Dummy variable = 1 consider as monthly income (Above 150,000 LKR) and 0 for otherwise | 4.1% |
| Civil Status (Married) | Dummy variable = 1 consider as civil status (Married) and 0 for otherwise | 56.7% |

Source: Authors' calculations.

researcher used reliability to clarify the questionnaire's ability to produce quality information for the analysis. The most critical element is the reliability of the questionnaire or data, as it underpins the reliability of the findings, which affects the conclusion of the study. Internal quality reliability is required since survey data is used for research. Cronbach's alpha value identified as the most common measure of reliability was used to assess if the internal instruments were reliable [43]. Test-retest, alternate or parallel types, split-half procedure, and internal consistency are some of the techniques that can perform a questionnaire reliability test. Internal consistency is the most popular approach for determining the reliability of the questionnaire reliability, which was also utilized in this analysis. Cronbach's alpha value was used by the researcher to examine the same characteristics. A reliability coefficient greater than 0.6 suggests a high level of accuracy. The study used Cronbach's alpha to assess the internal consistency or the degree to which objects quantify different aspects of the same definition. A value greater than or equal to 0.6 was considered appropriate [43]. These measurements were accepted as appropriate to explain the findings of this analysis.

Furthermore, the test indicates that if any items were excluded from the test, the Cronbach alpha reliability would not fall below the acceptable value of 0.60. As a result, when an object with an aggregate rating of 0.60 is discarded, all dimensions shown in Table 2 indicate a high value of reliability. This indicates that all elements of the questionnaire are important to indicate and that removing any one of these would result in a lower Cronbach Alpha, a sign of poor reliability. Table 3 summarises the participants' views on all independent and dependent variables considered in this study.

The mean value of 3.2703 indicates the variable on Work-life balance is 3.2703 and the mode value 3.4444 explains the agreement of the participants related to each question item. It explains the different attitudes of the participants related to work-life balance. The standard deviation small as 0.3442 means most respondents have a similar view as the mean value and it shows that data set not gone beyond the mean value and data set keeps around the mean value. The skewness of the data with -0.4734 means that data does not significantly differ from normality. The kurtosis value 10.6769 (greater than 3) means leptokurtic, which explains the deviation from a normal distribution. Confidence intervals were checked and confident levesl were found to be low; these are presented in S3 Appendix.

For the working environment variable, the mean value of 3.5726 and mode value 3.4444 explain the agreement of the participants related to each question item. It means the working environment is an impact on the work-life balance. Standard deviation small as 0.2289 means a minor variation from the mean value; thus, the feedback of most participants resembles similar views. The skewness of the data 1.9231 means that negatively skewed data are concentrated on the left side of the tale. The kurtosis value greater than 3 (value 15.8525) means a leptokurtic style of data distribution; as noted previously, describes the deviation from the normal distribution.

**Table 2. Cronbach's alpha value–on independent variables.**

| Variable | Reliability (Cronbach's alpha) | Scale Variance if Item Deleted | Corrected Item-Total Correlation | Squared Multiple Correlation | Cronbach's Alpha if Item Deleted | Number of question items |
|---|---|---|---|---|---|---|
| WE | 0.7495 | 0.6555 | 0.5281 | 0.2965 | 0.6528 | 18 No's |
| NWE | 0.7270 | 0.2506 | 0.5800 | 0.3376 | 0.6358 | 05 No's |
| Overall Cronbach's Alpha | 0.6802 | | | | | |

Source: Authors' calculations

**Table 3. Descriptive statistics of independent and dependent variables.**

|  | Work-life balance | Working Environment | Non-working Environment |
|---|---|---|---|
| N | 270 | 270 | 270 |
| Mean | 3.2703 | 3.5726 | 3.3637 |
| Std. Error of Mean | 0.0209 | 0.0139 | .0345 |
| Median | 3.5556 | 3.5556 | 3.0000 |
| Mode | 3.4444 | 3.4444 | 3.0000 |
| Std. Deviation | 0.3442 | 0.2289 | 0.5680 |
| Variance | 0.1185 | 0.0524 | 0.3226 |
| Skewness | -0.4734 | -1.9231 | 0.8914 |
| Std. Error of Skewness | 0.1482 | 0.1482 | 0.1482 |
| Kurtosis | 10.6769 | 15.8525 | 1.6158 |
| Std. Error of Kurtosis | 0.2954 | 0.2954 | 0.2954 |

Source: Authors' calculations.

Moreover, regarding participants' views related to the non-working environment variable, the mean value of 3.3637 and mode 3.00 explained the agreement of the participants related to each question item. As in the previous variable working environment, the standard deviation small as 0.5680 means viewpoints of most respondents do not significantly vary from that of others. The skewness of the data -0.8914 means positively skewed data concentrated on the right side of the tale. The kurtosis value of 1.6158 (less than 3) indicates a platykurtic style of data distribution. Table 4 shows a general description of the demographics of the respondents.

## Regression results

If the independent variable is known, the dependent variable can be estimated using regression analysis. This test can be used to identify and assess variables that affect the most on employee results. The regression analysis output is classified into three models. Since there are two independent variables in this analysis, the study used multiple regression and the forward stepwise regression technique. New variables for selection with p-value <0.5 and previously selected variables for removal with p-value ≥0.10. Gender, education level, monthly income, and civil status variables were not selected by the stepwise technique.

To ensure that the independent variables were not strongly correlated to each other, multicollinearity was tested using the variance inflation factor (VIF) and tolerance. A forward stepwise technique helped pick variables in each specification. The reliability of the findings was assessed using three separate model diagnostic criteria. According to the forward stepwise approach, adding variables did not affect the importance of the existing variables. The VIF and tolerance were also measured and found to be low (Tolerance maximum level .9946 and VIF maximum level 1.2552), suggesting that multicollinearity is not significant in this study.

The statistical method of regression analysis is used to investigate the relationship between two or more variables of interest. However, despite many different forms of regression, all these methods have a common goal: i.e. to examine the relationship between one or more independent variables and a dependent variable. Working environment, non-working environment, and dummy variables (Age, Gender, Education level, Monthly income, and civil states responsiveness) act as predicting variables in the present study, while work-life balance is the outcome variable. Table 5 shows that all the independent variables in the standard model, including working environment and non-working environment substantially represent work-life balance.

**Table 4. General description of the demographics.**

| Demographics | Categories | N | Presentage (%) |
|---|---|---|---|
| **Age** | 20–25 Years | 34 | 12.6 |
| | 26–30 years | 106 | 39.3 |
| | 31–40 years | 102 | 37.8 |
| | 41–50 years | 28 | 10.4 |
| **Gender** | Male | 130 | 48.1 |
| | Female | 140 | 51.9 |
| **Education Level** | GCE A/L | 9 | 3.3 |
| | Diploma | 38 | 14.1 |
| | Degree | 139 | 51.5 |
| | Under Graduate | 74 | 27.4 |
| | Post Graduate | 10 | 3.7 |
| **Monthly Income** | Below 50,000 | 23 | 8.5 |
| | 50,000–100,000 | 169 | 62.6 |
| | 100,000–150,000 | 67 | 24.8 |
| | Above 150,000 | 11 | 4.1 |
| **Civil status** | Single | 117 | 43.3 |
| | Married | 153 | 56.7 |
| **If married, does the spouse do a job** | Yes | 118 | 43.7 |
| | No | 81 | 30.0 |
| | Not answered | 71 | 26.3 |
| **No's of children** | Yes | 130 | 48.1 |
| | No | 69 | 25.6 |
| | Not answered | 71 | 26.3 |
| **Working platform** | Virtual office platform | 107 | 39.6 |
| | Traditional office platform | 0 | 0 |
| | Both platforms | 163 | 60.4 |
| **virtual office hours longer than your regular working hours** | Yes, most of the time | 125 | 46.3 |
| | No, Never | 15 | 5.6 |
| | Yes, rarely | 130 | 48.1 |
| **Working on weekends** | Yes, most of the time | 132 | 48.9 |
| | No, Never | 13 | 4.8 |
| | Yes, rarely | 125 | 46.3 |
| **Spending worth time with family** | Yes | 225 | 83.3 |
| | No | 5 | 1.9 |
| | Sometimes | 40 | 14.8 |
| **Total** | | **270** | **100.0** |

Source: Authors' calculations.

## Discussion

### Gender and no of children towards work life balance

In the first model (1), significant positive signs of non-working environment dimensions (Gender and number of children) indicate a positive and significant influence of the non-working-environment towards the work-Life balance. This aligns with the objective of the study, which is to identify the impact of the virtual office platform on work-life balance in the Sri Lankan context. The positive coefficient of the non-working environment in regression

**Table 5. Regression results-on variables.**

| Variable | Model 1 | Model 2 | Model 3 |
|---|---|---|---|
| **Working environment** | 0.3264*** | 0.3168*** | 0.3219*** |
| **Non-Working Environment Age** | 0.4010*** | 0.4105*** | 0.4098*** |
| 20–25 | | 0.0064* | |
| 26–30 | | -0.1102*** | |
| 31–40 | | | 0.1078*** |
| 41–50 | | -0.0801*** | |
| **Gender** | | | |
| Female | | 0.0043* | |
| **Education Level** | | | |
| GCE (A/L) | | 0.0434** | |
| Diploma | | 0.0453** | |
| Degree | | 0.0650** | |
| Post Graduate | | 0.0165* | |
| **Monthly Income (LKR)** | | | |
| Below 50,000 | | -0.0611** | |
| 100,000–150,000 | | 0.0787*** | |
| Above 150,000 | | 0.0333** | |
| **Civil Status** | | | |
| Married | | 0.0110* | |
| **Constant** | 0.6994* | 0.7219* | 0.6765*** |
| **R-Squared** | 0.3842 | 0.4042 | 0.3958 |
| **Adj. R-Squared** | 0.3796 | 0.3715 | 0.3890 |
| **Std. Error of the Estimate** | 0.2711 | 0.2729 | 0.2690 |
| **Observation** | 269 | 269 | 269 |

Note

*Significant at 10% level

**Significant at 5% level

***Significant at 1% level.

Source: Authors' calculations.

results table shows that 'Gender and number of children will mainly impact the work-life balance in a virtual platform. To be more precise, there would be a 0.40% effect on work-life balance for every percentage point change in the non-working environment. This means that mean non-working environment (Gender and no of children) factors will mainly affect work life balance in 40%. Similarly, the positive coefficients for the working environment indicate that for every percentage improvement in each item, the effect on work-life balance is 0.32%; both results are highly significant.

## The working environment and the non-working environment on work life balance

Since there is a strong and positive association between predicting and outcome variables, the significance level of 'working environment' and 'non-working environment' in model two (2) means that one percentage point change in each item will result in 0.31% and 0.41% impact on work-life balance, respectively. Similar to model one (1), model two (2) also confirms that there is a positive and significant influence of the non-working-environment towards the work-life balance, which aligns with the objective of the study.

Moreover, for better results, dummy variables were used in model 2 (Age, Gender, Education level, Monthly income, and civil status). The positive coefficient of the 'Working environment' in model two (2) revealed that a one percentage point effect results in a 0.31% work-life balance. Every percentage point shift in the non-working environment impact 0.41% on work-life balance. The results suggest that every employee, as well as the employer who carried out official tasks via a virtual platform, needs to pay careful attention to non-work-environment factors. Additionally, regression results table shows that dummy variables highlight those in 26–30 age groups having more significant impacts related to virtual office platforms on their WLB. Increasingly, people are contributing remotely via online platforms for multiple employers in various roles as part-time or freelance workers. Management must carefully identify these factors and their impacts when they formulate future policies and incorporate these into their companies work practices, especially considering this 26–30 age group with significant impacts on WLB. Also, regression results table shows that females having a more favorable impact on virtual office on their work-life balance compared to males. This is widely evident as virtual platforms enabled female employees to switch between personal and career roles while working in homes. When considering the impact of education level on work-life balance outcomes, degree holders seem to have more impact in contrast to other levels of education considered in this study. Likewise, based on results, the level of income effect on work-life balance affect with the LKR 100,000–150,000 income group having more impact on work-life balance. Furthermore, in terms of marital status, the married category has more impacts on work-life balance as per model two (2).

To better evaluate, model three (3) was created by using the forward stepwise regression technique. In regression results table, there are similar indications in models one (1) and two (2). Model three (3) indicated that the non-working environment factors increasing by 1% may increase impacts by 41% in work-life balance and working environment factors increasing by 1 percentage point may increase impacts by 32% in work-life balance. Moreover, the model highlights an increase in 1 one percentage point in 31–40 age group factors having a 10% impact on their work-life balance. This implies that employees in 31–40 age group have more work life impacts compared to those of other age groups while they carry out work on a virtual platform. R square value of 0.3958 means that 39.58% variance of work-life balance can be predicted using all the independent variables, namely working environment and non-working environment. Simpy, this implies that 39.58% of changes in work life balance happen due to the factors related to working environment and the non-working environment. All models highlighted that there is a positive and significant influence of the working environment and non-working-environment towards the work-life balance. Among these, the non-working environment has a greater impact on work-life balance. Accordingly, the study results justify the objective of the study. Overall, three (03) models prove all hypotheses support $H_1$ and $H_2$, whereas $H1_0$ and $H2_0$ are rejected. As such, this study supports the previous researches findings [8, 14, 23, 28].

## Working from a virtual platform–Work life balance and workplace performance of employees

According to the experimental evidence presented previously, the virtual office platform positively affects work-life balance. The results corroborate the findings of previous studies. Tietze, Musson [38] identified through systematic keyword searches of multi-disciplinary databases and suggested that home-based work (Virtual office) or working remotely from an employer or from a conventional place of work for a long period, is a flexible way of working that encompasses a wide variety of work activities. As such, these findings mainly describe that

gender has a positive impact on work-life balance while working virtually. Fischlmayr and Kollinger [36] hold the viewpoint that workplace interference with the family has a greater effect on the expatriate's family's well-being. Mroeover, researchers discovered from the results that there is an issue of female expatriates and their work-life balance. Calvo-Salguero, Salinas [37] found out that there are no gender disparities in family interfering with job duties since the time spent on the family affects the degree to which women encounter the conflict. Under the non-working environment variable, gender and number of children have a direct link with virtual office performances as well as family life [41]. In the Sri Lankan context, Malalasekara [21] examined the relationship between work-life balance and the workplace performance of employees. Findings discovered that employee-related aspects of family roles may have a significant positive effect on the success of both private and government sector workers.

The scope of this study was limited in terms of the sample size and geographical coverage. The study findings and analyses were restricted to the two districts (Colombo and Kandy), with a total of 270 virtual platform users and the use of online questioners through Google Forms. It is reasonable to assume that demographics in terms of teleworking workers can be similar, yet distinctive from the remaining districts in Sri Lanka. These districts, hence, the generalizability of the findings has some limitations. Therefore, future research can consider extending the scope of the study countrywide, to include a large number and various types of virtual users in Sri Lanka.

In general, all three models taken into consideration point out that there is a significant positive impact of the virtual office platform on work-life balance. Impacts are more inclined towards non-working environment factors (Gender and number of children). Concerning the effect of virtual offices on work-life balance, the study discovered that excess information, related publications, and vague information from the viewpoint of information providers contribute to three types of confusion, namely overload, similarity, and uncertainty. Furthermore, results comply with the main objective of identifying the impact of the virtual office platform on work-life balance in the Sri Lankan context. As a result, the present research affirms results of previous research carried out globally as well as those published.

## Conclusion

The primary goal of this research was to determine the impact of a virtual office platform on work-life balance in Sri Lanka. While several models have been used to investigate various aspects of virtual office platforms and their impact on work-life balance, only a few empirical studies have been conducted in the Sri Lankan context. Apart from this, the virtual office platform is relatively new and limited in the Sri Lankan working environment until the pandemic necessitated the transformation of conventional work patterns for businesses and the economy to stay afloat. In light of the above, this research establishes a baseline that can be considered for determining the impact of the virtual office platform on work-life balance. The study's numerical results revealed that work-life balance is linked to both the working and non-working environments. However, it should be highlighted that both the working and non-working environments are worth considering to identify the impact of virtual office platforms on employee work-life balance. Nevertheless, according to the analysis of the present study, relatively a higher weightage can be assigned for factors associated with non-working environments. Expanding the scope of the study in future researches incorporating factors relating to these can help discover insights unique to Sri Lanka to better address the issues, in this scope of research.

The attitude of employers' cultures in the virtual platform and policies have an impact on their employees' personal lives as well as their productivity according to the findings of the

study. Employees would be more productive if they believe their company trusts, recognizes and cares for them, and if they have been duly guided for preparation (including online processes, etc.), project management, and support to complete their tasks productively. When workers do not need to spend time, money, or resources on transportation between home and work, these have a positive impact on employee productivity. Apart from this, management must ensure that workers have or provided with minimum resources (tech devices such as laptops, internet facilities, etc.), a good working atmosphere (such as space, free from noise and disturbances at home, etc.), and a 'family friendly' culture to operate from home.

Based on the three models, the study also revealed that there is a significant positive impact of the virtual office platform on work-life balance. When working with virtual platforms, both the employee and the employer need to consider these variables carefully. In the context of efforts to reduce impacts linked to virtual platforms, some lessons can be learned from this Sri Lankan case study. Findings demonstrate that non-working environment factors which include gender and the number of children in a family have a major effect, particularly in developing countries like Sri Lanka. Furthermore, findings show that these effects were particularly strong in 26–30 and 31–40 age groups. As discussed previously, the gig economy growing at a remarkable rate and their income levels are generally high. In addition, the virtual platform has created opportunities for more female workers to rejoin the workforce by teleworking and also for those who have temporarily stayed away from work (due to being married, with kids, etc.) to return. In Sri Lanka, virtual platforms and teleworking need to be perceived beyond a mere quick fix to sustain the economy. These teleworking policies can be incorporated into organizational practices and processes. In this approach, much attention is to be paid for women with kids aged 0–5 years, skilled employees and degree holders (based on this study), and especially those who have the potential to adapt to working via online platforms. Overall, flexibility and more opportunities for workers, a proactive approach to address the role conflict which strongly impacts well-being in virtual platforms (where employees handle both work and personal roles, allow quality time with family) can help achieve positive organizational outcomes in the long run.

The results could be relevant and beneficial for a developing country like Sri Lanka to formulate policies, set up corporate strategies and processes, where virtual platforms are booming and contribute significantly to growth of the macro-economy. Meanwhile, in conclusion, this study contributes to the existing research gap relate to the impact of virtual working platforms related to work-life balance in the Sri Lankan context.

## Limitations

There are various flaws in this study. This study mainly focuses on linear impacts. Hence, not focusing the non-linear impacts is a main limitation of this study. Moreover, the study findings and analysis were limited only to two districts, with a total of 270 virtual platform users and online questioners using Google Forms. As a result, future research in Sri Lanka may expand this study on sample population and geographical area, thus involving a large number of virtual participants covering the entire country.

## Supporting information

**S1 Appendix. Questionnaire.**
(DOCX)

**S2 Appendix. Data file.**
(DOCX)

**S3 Appendix. Coefficients table and model summary for model 03.**
(DOCX)

**S1 Data.**
(SAV)

## Author Contributions

**Conceptualization:** Dharshana Rathnaweera, Ruwan Jayathilaka.

**Data curation:** Dharshana Rathnaweera.

**Formal analysis:** Dharshana Rathnaweera, Ruwan Jayathilaka.

**Investigation:** Ruwan Jayathilaka.

**Methodology:** Ruwan Jayathilaka.

**Software:** Dharshana Rathnaweera, Ruwan Jayathilaka.

**Supervision:** Ruwan Jayathilaka.

**Validation:** Dharshana Rathnaweera, Ruwan Jayathilaka.

**Visualization:** Ruwan Jayathilaka.

**Writing – original draft:** Dharshana Rathnaweera, Ruwan Jayathilaka.

**Writing – review & editing:** Ruwan Jayathilaka.

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
