## [Decision Letter · Decision Letter 0]

20 Sep 2021

PONE-D-21-14212In Employees’ Favour or Not? - The Impact of Virtual Office Platform on the Work-Life BalancesPLOS ONE

Dear Dr. Jayathilaka,

Thank you for submitting your manuscript to PLOS ONE. After careful consideration, we feel that it has merit but does not fully meet PLOS ONE’s publication criteria as it currently stands. Therefore, we invite you to submit a revised version of the manuscript that addresses the points raised during the review process.

Your manuscript has undergone the peer-review process and the reviewers have provided their comments/suggestions. Kindly address these points/concerns before we make a decision.

We look forward to receiving your revised manuscript.

Kind regards,

Kingston Rajiah

Academic Editor

PLOS ONE

Journal Requirements:

3. Please ensure that you refer to Figure 1 in your text as, if accepted, production will need this reference to link the reader to the figure.

4. We note you have included a table to which you do not refer in the text of your manuscript. Please ensure that you refer to Table 2 in your text; if accepted, production will need this reference to link the reader to the Table.

Reviewers' comments:

Reviewer's Responses to Questions

**Comments to the Author**

1. Is the manuscript technically sound, and do the data support the conclusions?

Reviewer #1: Partly

Reviewer #2: Partly

2. Has the statistical analysis been performed appropriately and rigorously? 

Reviewer #1: Yes

Reviewer #2: Yes

3. Have the authors made all data underlying the findings in their manuscript fully available?

Reviewer #1: No

Reviewer #2: No

4. Is the manuscript presented in an intelligible fashion and written in standard English?

Reviewer #1: No

Reviewer #2: Yes

5. Review Comments to the Author

Reviewer #1: PLOS ONE

Manuscript Number: PONE-D-21-14212

In Employees’ Favour or Not? – The Impact of Virtual Office Platform on the Work-Life Balances

First impressions of the study

A timely study has practical implications for working contexts that employ virtual office platforms and work from home strategies brought forward by the COVID-19 pandemic.

Manuscript - Lines need to be lined and numbered to provide specific comments line by line.

Structure – Requires correct structuring to,

a. Introduction

b. Literature review

c. Methodology

d. Results

e. Discussion

f. Conclusions

Title, Abstract, References:

Only ten out of 31 are from past the period of 2010 within the listed references. Most references are from the early 1990s. Multiple recent research publications on gender, work-life balance in the virtual environments are available on research databases, however not included in the study. See examples of some such below (web links provided),

References – recent study findings on work-life balance as references,

Bangladesh-

https://onlinelibrary.wiley.com/doi/full/10.1111/issj.12267

Italy –

https://papers.ssrn.com/sol3/papers.cfm?abstract_id=3644817

Indonesia-

http://journal.lembagakita.org/index.php/IJMSIT/article/view/231/218

Iceland-

https://onlinelibrary.wiley.com/doi/full/10.1111/gwao.12552

Introduction:

Overall, writing has to be reviewed for objectivity & scientific writing style, grammar, brevity, and logical flow. Missing references are observed on multiple occasions when facts are mentioned throughout the paper.

E.g., Page three-line 1,2 ‘VUCA’ comment – missing reference. Line 5 mentions a country but is not specified.

Page 3, paragraph 2 – Discussion on how WFH has impacted worker’s in Sri Lanka both negative and positively depending on the home environment. Although mentioned, the reference is missing.

‘Generation ‘X’’ comment – is missing a reference.

‘Virtual office’ comment – Specify device examples and provide a reference.

Page 4, paragraph 3 – “Studies have found..” comment – However, specific studies are not indicated, and references are not provided in the text.

What is current and already known about the topic needs to be strengthened within this section, stating what is known from previous research on WFH and work-life balance globally. This section needs to be revised in general for clarity.

Objectives and Problem statement:

Problem statement -

Repeats content from introduction, problem statement already providing results before data collection and analysis. E.g., women, especially women with kids, are affected by WFH. References are missing in the justification of the comment. Clear writing of the problem at hand is required.

The problem statement also discusses women with privilege and advantage and demonstrates a disconnect to introduction and study.

Objective –

The objective includes a discussion of the study and future implications of research noted as objectives. E.g., the third objective, influencing policymakers.

Needs revision to indicate the study-specific objectives in clear and direct sentences.

Literature review:

This section begins by stating that there are 59 publications included in the present study. However, the reference list only indicates 31.

Crossed out words present in the literature review body and writing styles differ from the introduction to the literature review.

Work-life balance

The work-life balance presented in the literature review is not adequately defined as per the requirement of this study. E.g., what is work-life balance? How is this defined and operationalized? How can work-life balance be measured? What are study findings of factors that influence work-life balance?

Working environment

The presentation of facts in this section is confusing to the reader. E.g., the ‘Working environment’ topic begins and proceeds with defining family conflict. Requires to be revised for logical presentation.

Additionally, Teleworking from the 1990s can be vastly different from the current virtual WFH experience. Are the references presented from 1992 relevant? Only provide the relevant information pertinent to the study at hand.

The literature review section has a feeling of having been a part of another writing task. It provides an unnecessary elaboration of facts that are not always useful to the reader. A summary with only the most significant and relevant studies should be mentioned within the paper.

Non- working environment

The writing as the paragraph begins has no logical connection to the provided subtopic. The logical flow is required to increase clarity, flow, and ease of readability. E.g., gender is being spoken of, although gender is not the topic without a connection to what is being presented, which is a ‘non-working environment.’

Methodology:

Spelling mistakes were observed within this section. Questions for the authors to clarify at this juncture include,

a. How were the institutes selected?

b. Can online surveys be “random sampling” as stated within the manuscript?

c. How was the sample selection ‘free of bias’?

d. What were the questionnaires used?

e. Who developed them? Are they made available as appendices to reviewers?

f. How is the validity of the questionnaires established?

g. Consistency reliability alone at times is insufficient. Was the measure taken through any other methods of rigorous reliability testing? Further, alpha values less than seven can be interpreted as ‘questionable’ (George & Mallery, 2003).

h. How was the sample size calculated?

i. How many questions were there? Replicability of study is not possible with the provided information.

Further comments,

Methodology sections are written in present tense “questionnaire intends to obtain..” which does not tally as study has already been conducted.

Further, the methodology section is observed to discuss results analysis and information relevant to results interpretation, which is irrelevant to this section.

The reference provided by Ben Akiva & Bowman is a study on residential location and activity and not a statistical reference as intended.

Overall, unclear how the conceptual model was operationalized.

Results:

“Data topic” appears to discuss methodology while “methodology” as a topic includes data analysis methods, not the study methodology per se. Logical topic selection and content are required for the ease of readability of the paper.

Page 18 – number of children not observed in table 4 as a variable.

Why is the conceptual model presented within methodology? Suggest it to be provided following the literature review to demonstrate the hypothesis derived from literature.

Results and discussion of results should be listed separately.

Page 19 – gig economy is on the rise is not related to the results section or result interpretation

Multicollinearity, VIF, stepwise regression tables are not presented, although mentioned under results as tested.

Results are discussed without reference to tables at times and unclear to the reader at times.

Particular text within this section is a repetition from other previous areas of the article.

Page 14 – demographic data is suggested to be presented as a table.

H1 H2 is mentioned as supported by results but also rejected. This statement requires clarification.

Indicate effect and confidence intervals in results to interpret practical meaningfulness.

Further, regression results alone do not clarify the impact of work environment factors and non-work environment factors on work-life balance in a meaningful manner. Linear regression only indicates that there is a positive relationship between the variables. The identified factors individually require to be analyzed for impact on work-life balance to conclude with further rigorous results analysis.

Discussion & Conclusion:

Spelling errors are observed and practical interpretation of the results needed.

Wording – ‘significant positive impact’ indicates a positive impact, but it only denotes a significant relationship between the two variables. Revision for clear discussion of results correct use of terminology in encouraged.

References not presented in lit review are apparent in the discussion.

Adding a ‘future implications’ and ‘limitations’ sections would be beneficial to paper.

Conclusions, need to be revised for further clarity.

Overall comments:

Requires major corrections to revise, re-write and streamline article into an article worthy of publication. Writing is disconnected and does not follow a logical flow within the report. Provided topics need to be logically positioned with relevant information discussed in brevity with the scientific writing style. The methodology does not provide adequate information for replicability with lacking measure development information. Study data are not available to reviewers. Results analysis and discussion, conclusions sections require further improvement.

References

George, D., & Mallery, P. (2003). SPSS for Windows step by step: A simple guide and reference (4th ed., Vol. 11.0). Boston: Allyn & Bacon.

Reviewer #2: The paper “In Employee’s Favour or Not? – The Impact of Virtual Office Platform on the Work-Life by Dharshana Rathnaweera,Ruwan Jayathilaka looks at observing the impact of virtual office platforms on work-life balance of people in Sri Lanka. Virtual offices are our adaptive response to the present pandemic and hence an area of interest for all (employee, employer, researchers) making the results of the present study beneficial.

Overall: It’s an interesting read. The strength of the paper lies in its contribution towards limited literature in this newly developed area of work, especially for the corresponding country. It would have benefitted the study more if some data could have been gathered from representative population of most geographical regions of the country in order to increase generalisation of results.

Abstract: A structured abstract with reasonable focus on the key areas. It would be nice to include more results in the abstract section as well. The author mentioned only one-line i.e “The results ascertain that both working and non-working environments are highly significant in impacting the work-life balance” in the result section of abstract. Suggest adding the key findings of the study in the result section of abstract. The current line added in the result section even do not explain the impact.

Introduction: Introduction is elaborate and extensive and well explanatory but it would be nice to support statements in introduction with current literature referencing as to make a stronger base. It will also be nice to elaborate about the ways as to ‘how the virtual office setup will help to recover from the economic setback’ as mentioned by the author.

Extensive review of literature has been carried out in the literature review section by the author to identify the independent and dependent variables for the study.

Methods: A planned and sufficiently designed methodology. Reliability of the questions has been checked with the help of internal consistency but it would have been nice to include another method (a for example a test-restest method) to increase reliability also along with checking Cronbach alpha. Also conducting a pilot study to check the validity of the questions would have helped. Also some increase in the sample size to avoid non response and self reporting bias could be given out as a recommendation for future studies. However the author’s effort to deduce results based on all three models really gives credibility to the responses.

Results and Discussion: The results section is not self-explanatory. It would be nice to include some information regarding the question asked to the population (as a table or theory) and their response for better understanding and clarity (atleast few, if not all) of the reader. Also it would have benefitted the paper if the results were explained simultaneously in a non technical fashion (for a layman) next to the technical aspects. The discussion is extensive and clearly identifies all factors that impact the work life balance in a working or a non working virtual setup.

Conclusion: Strongly recommend revising the conclusion section. The finding of this project identifies the impact and the factors that affect of a virtual office setup on work life balance. However, the author needs to revise the conclusion as currently it is very vague. Currently, the conclusion section describes the factors that affects work life balance and not the true impact on it.

6. PLOS authors have the option to publish the peer review history of their article (what does this mean?). If published, this will include your full peer review and any attached files.

Reviewer #1: No

Reviewer #2: No

---

## [Author Response · Author response to Decision Letter 0]

11 Oct 2021

Point–by–point response to reviewers

Comments from Authors: Please note that page numbers and line numbers refereed in this document is align with the manuscript which has track changes.

Comments of Reviewers:

Is the manuscript technically sound, and do the data support the conclusions?

Reviewer 01 – Partly

Reviewer 02 – Partly

Comments of Authors: Noted with thank you. This has been incorporated in the revised manuscript with track changes.

Comments of Reviewers:

Has the statistical analysis been performed appropriately and rigorously?

Reviewer 01 – Yes

Reviewer 02 – Yes

Comments of Authors:

Well noted. Thank you for your comment.

Comments of Reviewers:

Have the authors made all data underlying the findings in their manuscript fully available?

Reviewer 01 – No

Reviewer 02 – No

Comments of Authors:

Thank you comment has been noted. Data cannot be shared publicly because of data protection. Data are available from the Sri Lanka Institute of Information Technology, SLIIT Business School, Sri Lanka (contact via gazrat4ever@gmail.com or ruwan.j@sliit.lk ) for researchers who meet the criteria for access to confidential data. However, for the reviews purposes, we are sending the data file as a supporting document.

Comments of Reviewers:

Is the manuscript presented in an intelligible fashion and written in standard English?

Reviewer 01 – No

Reviewer 02 – Yes

Comments of Authors:

Thank you and well noted. This has been in cooperated in the revised manuscript with track changes.

5. Review Comments to the Author

Reviewer 1

Comments of Reviewers:

First impressions of the study

A timely study has practical implications for working contexts that employ virtual office platforms and work from home strategies brought forward by the COVID-19 pandemic.

Manuscript - Lines need to be lined and numbered to provide specific comments line by line.

Structure – Requires correct structuring to,

a. Introduction

b. Literature review

c. Methodology

d. Results

e. Discussion

f. Conclusions

Comments of Authors:

Thank you. Well, noted. As per the reviewer 1 suggestions have been corrected in the revised manuscript.

Comments of Reviewers:

Title, Abstract, References:

Only ten out of 31 are from past the period of 2010 within the listed references. Most references are from the early 1990s. Multiple recent research publications on gender, work-life balance in the virtual environments are available on research databases, however not included in the study.

Comments of Authors:

Comment has been noted and this has been corrected in the revised manuscript. Following 13 references has been included.

[6] (Page 4 - Line 2)

[2] (Page 2 - Line 28)

[15] (Page 7 - Line 19)

[16] (Page 7 - Line 19)

[11, 12] (Page 4 - Line 32)

[13, 14] (Page 5 - Line 13)

[15] (Page 7 - Line 18)

[19] (Page 8 - Line 1)

[8] (Page 8 - Line 1)

[22, 23] (Page 8 - Line 21)

[24] (Page 9 - Line 1)

[29] (Page 10 - Line 28)

[31] (Page 11 - Line 6)

Comments of Reviewers:

Introduction:

E.g., Page three-line 1,2 ‘VUCA’ comment – missing reference. Line 5 mentions a country but is not specified

Comments of Authors:

Thank you for your comment. However (short form of “Volatility, Uncertainty, Complexity, Ambiguity”. This has been corrected in revised version. 

(Page 3 line 10-11) and (Page 3 line 14)

Comments of Reviewers:

Page 3, paragraph 2 – Discussion on how WFH has impacted worker’s in Sri Lanka both negative and positively depending on the home environment. Although mentioned, the reference is missing.

Comments of Authors:

Comment is well noted. Reference has been added as suggested by reviewer 1. 

(Page 4 - line 1) 

Comments of Reviewers:

§ ‘Generation ‘X’’ comment – is missing a reference.

Comments of Authors:

Thank you for your comment. Reference has been added. 

(Page 4 - line 2)

Comments of Reviewers:

§ ‘Virtual office’ comment – Specify device examples and provide a reference.

Comments of Authors:

Comment is noted. Reference has been added as suggested by reviewer 1. 

(Page 4 line 4)

Comments 

of Reviewers:

§ Page 4, paragraph 3 – “Studies have found.” comment – However, specific studies are not indicated, and references are not provided in the text.

Comments of Authors:

Comment has been noted and this has been corrected in the revised manuscript. 

(Page 4 - line 32)

Comments of Reviewers:

§ What is current and already known about the topic needs to be strengthened within this section, stating what is known from previous research on WFH and work-life balance globally. This section needs to be revised in general for clarity.

Comments of Authors:

Well noted. That correction has been incorporated in to the introduction section as follows.

“Crosbie and Moore (1) imply that another endeavor that has been advocated as a technique to improve the work–life balance is homeworking. Also, employees working from residence are better as those who desire to work from home, have higher levels of job satisfaction, and have a better work–life balance under a rigorous contractual arrangement than with a soft law commitment”. 

(Page 2 – line 24-28) 

Comments of Reviewers:

Objectives and Problem statement:

Problem statement –

§ Repeats content from introduction, problem statement already providing results before data collection and analysis. E.g., women, especially women with kids, are affected by WFH. References are missing in the justification of the comment. Clear writing of the problem at hand is required.

Comments of Authors:

Comment has been noted and this has been corrected in the revised manuscript with following references.

[13,14]

(Page 5 - line 13)

Comments of Reviewers:

§ The problem statement also discusses women with privilege and advantage and demonstrates a disconnect to introduction and study.

Comments of Authors:

Thank you well noted this has been corrected in the revised manuscript.

(Page 5 - line 15-16).

Comments of Reviewers:

Objective –

§ The objective includes a discussion of the study and future implications of research noted as objectives. E.g., the third objective, influencing policymakers

Comments of Authors:

Highly appreciate your valuable comment. That correction has been incorporated in to the section as follows.

“Third, there has been very limited prior research project mainly related to the Sri Lankan context that implying future implications and policymaking related to the COVID-19 pandemic situation”

(Page 6 - line 8-10)

Comments of Reviewers:

§ Needs revision to indicate the study-specific objectives in clear and direct sentences.

Comments of Authors:

Comment has been noted and this has been corrected in the revised manuscript with following references.

[13,14]

(Page 5 - line 13)

Comments of Reviewers:

§ The problem statement also discusses women with privilege and advantage and demonstrates a disconnect to introduction and study.

Comments of Authors:

Thank you well noted this has been corrected in the revised manuscript.

(Page 5 - line 15-16).

Comments of Reviewers:

Objective –

§ The objective includes a discussion of the study and future implications of research noted as objectives. E.g., the third objective, influencing policymakers.

Comments of Authors:

Highly appreciate your valuable comment. That correction has been incorporated in to the section as follows.

“Third, there has been very limited prior research project mainly related to the Sri Lankan context that implying future implications and policymaking related to the COVID-19 pandemic situation”

(Page 6 - line 8-10)

Comments of Reviewers:

§ Needs revision to indicate the study-specific objectives in clear and direct sentences.

Comments of Authors:

Thank you for your comment. This has been indicated in paper as follows.

“The main objective of this study is to identify the impact of the virtual office platform on work-life balance in the Sri Lankan context”

(Page 5 – line 23-24)

“First, the present study enables employees to identify the areas that need to be considered when moving into a virtual office platform”

(Page 5- line 30-31)

“Second, particularly in light of the COVID-19 pandemic, it is critical to investigate the effect of the virtual office network on employees”

“Finally, the results would be useful to various industries and policymakers in an endeavor to recover from economic setbacks and improve quality of life”

(Page 6 – line 3-4)

Moreover, third objectives have been added to paper as suggestion given by reviewer 1.

“Third, there has been very limited prior research project mainly related to the Sri Lankan context that implying future implications and policymaking related to the COVID-19 pandemic situation”

(Page 6 – line 8-10)

Comments of Reviewers:

Literature review:

§ This section begins by stating that there are 59 publications included in the present study. However, the reference list only indicates 31.

Comments of Authors:

Thank you for your comment. This has been corrected in the revised manuscript with track changes.

(Page 7 - line 2-14)

Comments of Reviewers:

§ Crossed out words present in the literature review body and writing styles differ from the introduction to the literature review.

Comments of Authors:

Noted. This has been corrected in the revised manuscript.

Comments of Reviewers:

Work-life balance

The work-life balance presented in the literature review is not adequately defined as per the requirement of this study. E.g., what is work-life balance? How is this defined and operationalized? How can work-life balance be measured? What are study findings of factors that influence work-life balance?

Comments of Authors:

Well noted. This has been corrected and added new references in the revised manuscript. 

(Pages 7 line 18-20, Page 7 line 27-28 and page 8 line 1-4

Comments of Reviewers:

Working environment

§ The presentation of facts in this section is confusing to the reader. E.g., the ‘Working environment’ topic begins and proceeds with defining family conflict. Requires to be revised for logical presentation.

Comments of Authors:

Thank you for your comment. Added new references as per suggestion given by reviewer 1. 

(Pages 8 - lines 19-24) 

Comments of Reviewers:

§ Additionally, Teleworking from the 1990s can be vastly different from the current virtual WFH experience. Are the references presented from 1992 relevant? Only provide the relevant information pertinent to the study at hand.

Comments of Authors:

Comment has been noted and this has been corrected in the revised manuscript with track changes. 

(Pages 8- line 19-24 and page 9 - line 1-3) 

Comments of Reviewers:

§ The literature review section has a feeling of having been a part of another writing task. It provides an unnecessary elaboration of facts that are not always useful to the reader. A summary with only the most significant and relevant studies should be mentioned within the paper.

Comments of Authors:

The comment is well noted. The section has been corrected in the revised manuscript.

Comments of Reviewers:

Non- working environment

The writing as the paragraph begins has no logical connection to the provided subtopic. The logical flow is required to increase clarity, flow, and ease of readability. E.g., gender is being spoken of, although gender is not the topic without a connection to what is being presented, which is a ‘non-working environment.’

Comments of Authors:

Thank you for your comment. This has been corrected in the revised manuscript with track changes.

(Page 10 - line 23-28 and page 11 - line 6-11)

Comments of Reviewers:

Methodology:

Spelling mistakes were observed within this section. Questions for the authors to clarify at this juncture include,

Comments of Authors:

Thank you for the comment. spelling mistakes have been corrected in the revised manuscript.

Comments of Reviewers:

a. How were the institutes selected?

Comments of Authors:

These institutes were selected after observed which institutes were operating telework during the Covid-19 situation. Especially during the data collecting period.

Comments of Reviewers:

b. Can online surveys be “random sampling” as stated within the manuscript?

Comments of Authors:

The online questioner was not published to the general public. Online questioner link was only sent to selected persons of mention institutes.

Comments of Reviewers:

c. How was the sample selection ‘free of bias’?

Comments of Authors:

Online questions were sent to selected persons mainly considering free of bias covering all levels of management as well as covering all employees. Hence random sampling is mainly done by considering equality. 

Comments of Reviewers:

d. What were the questionnaires used?

Comments of Authors:

Questionnaires consist of main two sections. The first section (Section A) intends to gather demographic factors with 9 questions. And second section (Section B) was used for the independent and dependent variables. It comprises 32 questions. The used questioner will be attaching to mail as supporting documents (S1 Appendix).

Comments of Reviewers:

e. Who developed them? Are they made available as appendices to reviewers?

Comments of Authors:

A self-develop questioner was used.

Comments of Reviewers:

f. How is the validity of the questionnaires established?

Comments of Authors:

Pilot survey of 30 respondents were conducted to determine whether the questions were clear, understandable, and logical order. This has been corrected in the revised version.

(Page 14- line 25-26)

Comments of Reviewers:

g. Consistency reliability alone at times is insufficient. Was the measure taken through any other methods of rigorous reliability testing? Further, alpha values less than seven can be interpreted as ‘questionable’ (George & Mallery, 2003).

Comments of Authors:

The internal consistency was measured using Cronbach's alpha in this analysis. Moreover, Descriptive Statistics of independent and dependent variables were also measured. Both Cronbach's alpha results and Descriptive Statistics of independent and dependent variables results will be attached as supporting documents (S3 Appendix).

Comments of Reviewers:

h. How was the sample size calculated?

Comments of Authors:

The sample size was selected using online sample size calculator goodcalculators ( https://goodcalculators.com/sample-size-calculator/ ). The margin of accepted error was taken as 5% and confidence level of 95%. The results indicated that sample size of at-least 270 would be required. This has been incorporated to the revised version

(Page 14 line 21-24)

Comments of Reviewers:

i. How many questions were there? Replicability of study is not possible with the provided information.

Comments of Authors:

A sent questionnaire will be attaching to the mail as supporting documents. (Questionnaires consist of main two sections. The first section (Section A) intends to gather demographic factors with 9 questions. And second section (Section B) was used for the variables. It comprises with 32 questions)

Comments of Reviewers:

Further comments

§ Methodology sections are written in present tense “questionnaire intends to obtain.” which does not tally as study has already been conducted.

Comments of Authors:

Thank you for your comment. This has been corrected in the revised manuscript with track changes (Page 15 - line 5-6)

Comments of Reviewers:

§ Further, the methodology section is observed to discuss results analysis and information relevant to results interpretation, which is irrelevant to this section.

Comments of Authors:

Comment is noted. Correction has been done in the revised manuscript.

(Page 14-line 14-31 and Page 15 – line 1-4)

Comments of Reviewers:

§ The reference provided by Ben Akiva & Bowman is a study on residential location and activity and not a statistical reference as intended.

Comments of Authors:

Deleted the mentioned references and added new references relevant to goodness-of-fit. (Page 20-line 8)

Comments of Reviewers:

§ Overall, unclear how the conceptual model was operationalized.

Comments of Authors:

The operationalization table which helps to create a conceptual model will be attaching to mail as supporting documents (S3 Appendix).

Comments of Reviewers:

Results:

§ “Data topic” appears to discuss methodology while “methodology” as a topic includes data analysis methods, not the study methodology per se. Logical topic selection and content are required for the ease of readability of the paper.

§ Page 18 – number of children not observed in table 4 as a variable.

Comments of Authors:

Thank you for your comment. This has been corrected in the revised manuscript with track changes.

(Page 16 - line 8-17 and page 20 line 12-18)

Comments of Reviewers:

§ Page 18 – number of children not observed in table 4 as a variable.

Comments of Authors:

Thank you very much for your comment. In the first model, significant positive signs are indicated under a non-working environment. On- working environment consists of gender and no of children as per the given conceptual model. Dummy variables were used in model two only selected demographic variables. (Age, Gender, Education level, Monthly income, and civil status). Model three was created by using the forward stepwise regression technique and table 5 indicated system selected variables only (Age group 31-40).

Moreover, all the three models have highlighted the positive influence of the working environment and non-working-environment on the work-life balance, and from that non-working environment has a significant influence (Gender and no of children).

Comments of Reviewers:

§ Why is the conceptual model presented 

within methodology? Suggest it to be provided following the literature review to demonstrate the hypothesis derived from literature.

Comments of Authors:

Thank you for your comment. As per suggestion given by reviewer 1 this has been moved later part of the literature review.

(Page 13 line 22-26 and page 14 line 1-11)

Comments of Reviewers:

§ Results and discussion of results should be listed separately.

Comments of Authors:

Well, noted. This has been corrected in the revised manuscript.

Comments of Reviewers:

§ Page 19 – gig economy is on the rise is not related to the results section or result interpretation

Comments of Authors:

Noted with thank you. This has been corrected in the revised manuscript with track changes. (Page 24 - line 4 and Page 25 – line 3-6)

Comments of Reviewers:

§ Multicollinearity, VIF, stepwise regression tables are not presented, although mentioned under results as tested.

Comments of Authors:

Comment is noted. Multicollinearity, VIF, stepwise regression tables will be attaching to mail as supporting documents (S3 Appendix).

(Page 25 - line 21)

Comments of Reviewers:

§ Results are discussed without reference to tables at times and unclear to the reader at times.

Comments of Authors:

Noted. This has been corrected with track changes. 

(Page 18-line 7) (Page 21-line 16) (Page 24-line 7) (Page 24-line 34) (Page 25-line 24)

Comments of Reviewers:

§ Particular text within this section is a repetition from other previous areas of the article.

Comments of Authors:

As per the authors knowledge, that has been corrected in the revised manuscript.

Comments of Reviewers:

§ Page 14 – demographic data is suggested to be presented as a table.

Comments of Authors:

Well noted. This has been corrected in the revised manuscript with track changes. 

(Page 18-line 20 and page 19)

Comments of Reviewers:

§ H1 H2 is mentioned as supported by results but also rejected. This statement requires clarification.

Comments of Authors:

The comment is well noted, and it has been corrected in the revised manuscript with track changes. 

(Page 25-line 29-31)

Comments of Reviewers:

§ Indicate effect and confidence intervals in results to interpret practical meaningfulness.

Comments of Authors:

Comment is will noted. This has been corrected as follows. “Confidence intervals was checked and presented in S3 appendix and found that confident level is low” 

(Page 22-line 10-11)

Comments of Reviewers:

§ Further, regression results alone do not clarify the impact of work environment factors and non-work environment factors on work-life balance in a meaningful manner. Linear regression only indicates that there is a positive relationship between the variables. The identified factors individually require to be analyzed for impact on work-life balance to conclude with further rigorous results analysis.

Comments of Authors:

Very much thank you for four comments. This has been added as limitation as follows. “This study mainly focusing in to linear impact only. Hence as a limitation study is not focusing the non- linear impacts”

(Page 29 - line 28-32)

Comments of Reviewers:

Discussion & Conclusion:

§ Spelling errors are observed and practical interpretation of the results needed.

Comments of Authors:

Thank you for your comment. This has been corrected in the revised manuscript.

Comments of Reviewers:

§ Wording – ‘significant positive impact’ indicates a positive impact, but it only denotes a significant relationship between the two variables. Revision for clear discussion of results correct use of terminology in encouraged.

Comments of Authors:

Comment is noted. All three models were highlighted that that working and non-working environments were significant positive impacts on the work-life balance and among them non-working environment has having most significant positive impacts on the work-life balance. That is why the author mentions ‘significant positive impact’ in the paper.

Comments of Reviewers:

§ References not presented in lit review are apparent in the discussion.

Comments of Authors:

Comment is well noted. This has been corrected in the revised manuscript.

(Page 8 line 1-4 and page 12 line 18-20)

Comments of Reviewers:

§ Adding a ‘future implications’ and ‘limitations’ sections would be beneficial to paper.

Comments of Authors:

Comment has been noted and this has been added to revised manuscript as follows.

“There are various flaws in this study. This study mainly focusing in to linear impact only. Hence as a limitation study is not focusing the non- linear impacts. Moreover, the study's findings and analysis were limited to only two districts, with a total of 270 virtual platform users and online questioners using Google Forms. As a result, future research in Sri Lanka may involve a large number of virtual participants covering the entire country.

(Page 29 line 28-32) 

Comments of Reviewers:

Conclusions, need to be revised for further clarity.

Comments of Authors:

Well, noted. This has been corrected in the revised manuscript.

Comments of Reviewers:

Overall comments:

Requires major corrections to revise, re-write and streamline article into an article worthy of publication. Writing is disconnected and does not follow a logical flow within the report. Provided topics need to be logically positioned with relevant information discussed in brevity with the scientific writing style. The methodology does not provide adequate information for replicability with lacking measure development information. Study data are not available to reviewers. Results analysis and discussion, conclusions sections require further improvement.

Comments of Authors:

Comment has been noted and this has been corrected in the revised manuscript. New appendixes and the data file were added to the revised manuscripts.

Reviewer 2

Comments of Reviewers:

The paper “In Employee’s Favour or Not? – The Impact of Virtual Office Platform on the Work-Life by Dharshana Rathnaweera,Ruwan Jayathilaka looks at observing the impact of virtual office platforms on work-life balance of people in Sri Lanka. Virtual offices are our adaptive response to the present pandemic and hence an area of interest for all (employee, employer, researchers) making the results of the present study beneficial

Comments of Authors:

Thank you very much for your comment.

Comments of Reviewers:

Overall:

It’s an interesting read. The strength of the paper lies in its contribution towards limited literature in this newly developed area of work, especially for the corresponding country. It would have benefitted the study more if some data could have been gathered from representative population of most geographical regions of the country in order to increase generalisation of results.

Comments of Authors:

Well noted. Thank you for your comment.

Comments of Reviewers:

Abstract: 

A structured abstract with reasonable focus on the key areas. It would be nice to include more results in the abstract section as well. The author mentioned only one-line i.e “The results ascertain that both working and non-working environments are highly significant in impacting the work-life balance” in the result section of abstract. Suggest adding the key findings of the study in the result section of abstract. The current line added in the result section even do not explain the impact.

Comments of Authors:

Comment has been noted and this has been corrected in the revised as follows. “The non-working environment, in particular, has a bigger influence on work balance (Gender and No of children)” 

(Page 2 - line 15-16)

Comments of Reviewers:

Introduction: 

Introduction is elaborate and extensive and well explanatory but it would be nice to support statements in introduction with current literature referencing as to make a stronger base. It will also be nice to elaborate about the ways as to ‘how the virtual office setup will help to recover from the economic setback’ as mentioned by the author.

Extensive review of literature has been carried out in the literature review section by the author to identify the independent and dependent variables for the study.

Comments of Authors:

Comment has been noted and the introduction part has been corrected in the revised manuscript. following 13 latest references has been added to revised manuscript.

[6] (Page 4 - Line 2)

[2] (Page 2 - Line 28)

[15] (Page 7 - Line 19)

[16] (Page 7 - Line 19)

[11, 12] (Page 4 - Line 32)

[13, 14] (Page 5 - Line 13)

[15] (Page 7 - Line 18)

[19] (Page 8 - Line 1)

[8] (Page 8 - Line 1)

[22, 23] (Page 8 - Line 21)

[24] (Page 9 - Line 1)

[29] (Page 10 - Line 28)

[31] (Page 11 - Line 6)

Comments of Reviewers:

Methods: 

A planned and sufficiently designed methodology. Reliability of the questions has been checked with the help of internal consistency but it would have been nice to include another method (a for example a test-restest method) to increase reliability also along with checking Cronbach alpha. Also conducting a pilot study to check the validity of the questions would have helped. Also some increase in the sample size to avoid non response and self reporting bias could be given out as a recommendation for future studies. However the author’s effort to deduce results based on all three models really gives credibility to the responses.

Comments of Authors:

Thank you very much for your comment. For better evaluation apart from internal consistency, model three (3) results gone with multicollinearity, VIF factors with confidence intervals. Pilot survey was conducted, and this has been correct in the revised version. 

Comments of Reviewers:

Results and Discussion:

The results section is not self-explanatory. It would be nice to include some information regarding the question asked to the population (as a table or theory) and their response for better understanding and clarity (atleast few, if not all) of the reader. Also it would have benefitted the paper if the results were explained simultaneously in a non technical fashion (for a layman) next to the technical aspects. The discussion is extensive and clearly identifies all factors that impact the work life balance in a working or a non working virtual setup.

Comments of Authors:

Thank you for your comment. Now the revise version incorporates to the questioner added as an S1 appendix due to the length of the questioner.

Also, correction has been done to revised manuscript.

(Page 22 line 10-11)

(Page 24 line 17-19)

(Page 25 – line 3-6 and line 33-34)

Comments of Reviewers:

Conclusion:

Strongly recommend revising the conclusion section. The finding of this project identifies the impact and the factors that affect of a virtual office setup on work life balance. However, the author needs to revise the conclusion as currently it is very vague. Currently, the conclusion section describes the factors that affects work life balance and not the true impact on it.

Comments of Authors:

Comment has been noted and this part has been corrected in the revised manuscript.

---

## [Decision Letter · Decision Letter 1]

26 Oct 2021

PONE-D-21-14212R1In Employees’ Favour or Not? - The Impact of Virtual Office Platform on the Work-Life BalancesPLOS ONE

Dear Dr. Jayathilaka,

Thank you for submitting your manuscript to PLOS ONE. After careful consideration, we feel that it has merit but does not fully meet PLOS ONE’s publication criteria as it currently stands. Therefore, we invite you to submit a revised version of the manuscript that addresses the points raised during the review process.

The reviewer has recommended further revisions. Kindly address these before we make a decision.

We look forward to receiving your revised manuscript.

Kind regards,

Kingston Rajiah

Academic Editor

PLOS ONE

Journal Requirements:

Reviewers' comments:

Reviewer's Responses to Questions

**Comments to the Author**

1. If the authors have adequately addressed your comments raised in a previous round of review and you feel that this manuscript is now acceptable for publication, you may indicate that here to bypass the “Comments to the Author” section, enter your conflict of interest statement in the “Confidential to Editor” section, and submit your "Accept" recommendation.

Reviewer #1: All comments have been addressed

Reviewer #2: All comments have been addressed

2. Is the manuscript technically sound, and do the data support the conclusions?

Reviewer #1: Yes

Reviewer #2: Partly

3. Has the statistical analysis been performed appropriately and rigorously? 

Reviewer #1: Yes

Reviewer #2: Yes

4. Have the authors made all data underlying the findings in their manuscript fully available?

Reviewer #1: No

Reviewer #2: No

5. Is the manuscript presented in an intelligible fashion and written in standard English?

Reviewer #1: Yes

Reviewer #2: Yes

6. Review Comments to the Author

Reviewer #1: Abstract

• Common practice to write in words any number under two digits – “9” as “nine”

• Review sentences for succinct delivery and brevity.

• Line 11 – research gap sentence to precede the objective of the study to improve flow.

• Suggestion - Line 17 and 18 does can be combined with the following sentence – E.g. “Findings indicate that it is vital that…….. xx xx so that best use of virtual work environments can be obtained”

• Combine line 14, 15, 16 to form one sentence

Introduction

-Overall comment - Review for succinct presentation of ideas and brevity of language. Synthesize and present ideas in summary to the reader removing all and any unncessary information/words (review for brevity). For E.g. Line 24 – “Crosbie and Moore state that homeworking or working from home can improve the work-life balance”.

Problem statement

• Problem statement is repetitive of introduction and includes information re: women and literature that should be provided with in the introduction/review section. Problems statement should remain a very clear statement of the problems without other information.

Objective of study

• This section also provides rationale/justification of study which should be provided separately, prior to objectives.

Literature review

Overall comment - Review for succinct presentation of ideas and brevity of language. Synthesize and present ideas in summary to the reader removing all and any unncessary information/words (review for brevity).

Methodology

• The first sentence repeats purpose of study - please remove.

• - Subtopic 'Data' can be removed with passage directly under method.

• Page 16, conceptual model diagram is repeated and must be removed.

• Hypothesis should logically be provided under the problem statement or within that section, and not under method.

• Methodology can include measure development information in it as part of it, and results containing all results including demographics from the study - please revise - confusing to reader.

• A discussion section is presented before analysis - the use of topics/subtopics are confusing to the reader.

• Why is reliability analysis of the tool provided after presenting demographic information of the study population? Please be mindful of the flow of the article.

• Clearly demark 'discussion' following results presentation and discuss all matters related to findings under that subtopic.

Consluions -

Insert limitations as separate topic

Review overall article for logical flow of presentation and language edit for brevity.

Reviewer #2: No further comments to add. The author mentioned that the Data cannot be shared publicly because of data protection.

7. PLOS authors have the option to publish the peer review history of their article (what does this mean?). If published, this will include your full peer review and any attached files.

Reviewer #1: No

Reviewer #2: No

---

## [Author Response · Author response to Decision Letter 1]

2 Nov 2021

Point–by–point response to reviewers

Comments from Authors: Please note that page numbers and line numbers refereed in this document is align with the revised manuscript which has track changes.

Comments of Reviewers: 

If the authors have adequately addressed your comments raised in a previous round of review and you feel that this manuscript is now acceptable for publication, you may indicate that here to bypass the “Comments to the Author” section, enter your conflict of interest statement in the “Confidential to Editor” section, and submit your "Accept" recommendation.

Reviewer #1: All comments have been addressed

Reviewer #2: All comments have been addressed

Comments of Authors: Well noted. Thank you for your comment.

Comments of Reviewers:

Is the manuscript technically sound, and do the data support the conclusions?

Reviewer #1: Yes

Reviewer #2: Partly

Comments of Authors: Thank you comment has been noted. 

Comments of Reviewers:

Has the statistical analysis been performed appropriately and rigorously?

Reviewer #1: Yes

Reviewer #2: Yes

Comments of Authors: Thank you very much for your comment.

Comments of Reviewers:

Have the authors made all data underlying the findings in their manuscript fully available?

Reviewer #1: No

Reviewer #2: No

Comments of Authors: Thank you comment has been noted. 

Data cannot be shared publicly because of data protection. Data are available from the Sri Lanka Institute of Information Technology, SLIIT Business School, Sri Lanka (contact via gazrat4ever@gmail.com or ruwan.j@sliit.lk ) for researchers who meet the criteria for access to confidential data.

Comments of Reviewers:

Is the manuscript presented in an intelligible fashion and written in standard English?

Reviewer #1: Yes

Reviewer #2: Yes

Comments of Authors: Thank you. Well noted.

Review Comments to the Author

Reviewer 1

Comments of Reviewers:

Abstract

Common practice to write in words any number under two digits – “9” as “nine”

Comments of Authors: Comment has been noted and this has been corrected in the revised manuscript. (Page 2 - line 5)

Comments of Reviewers:

Review sentences for succinct delivery and brevity.

Line 11 – research gap sentence to precede the objective of the study to improve flow.

Comments of Authors: This has been corrected in the revised version.

Thank you. Comment has been noted and this has been corrected in the revised manuscript (Page 2 - line 9-13)

Comments of Reviewers:

Suggestion - Line 17 and 18 does can be combined with the following sentence – E.g. “Findings indicate that it is vital that…….. xx xx so that best use of virtual work environments can be obtained”

Comments of Authors: Well noted and this has been corrected in the revised manuscript (Page 2 - line 18-21).

Comments of Reviewers:

Combine line 14, 15, 16 to form one sentence

Comments of Authors: Comment has been noted and this has been corrected in the revised manuscript (Page 2 - line 13-18).

Comments of Reviewers:

Introduction:

-Overall comment - Review for succinct presentation of ideas and brevity of language. Synthesize and present ideas in summary to the reader removing all and any unncessary information/words (review for brevity). For E.g. Line 24 – “Crosbie and Moore state that homeworking or working from home can improve the work-life balance”.

Comments of Authors: Comment has been well noted and this has been corrected in the revised manuscript (Page 2 - line 27-29).

Comments of Reviewers:

Problem statement:

Problem statement is repetitive of introduction and includes information re: women and literature that should be provided with in the introduction/review section. Problems statement should remain a very clear statement of the problems without other information.

Comments of Authors: Thank you for your comment. This has been corrected in the revised manuscript (Page 4 - line 31-34 and Page 5 – line 14-17).

Comments of Reviewers:

Objective of study:

This section also provides rationale/justification of study which should be provided separately, prior to objective

Comments of Authors: Comment is well noted and this has been corrected in the revised manuscript. Rational of the study has been stated before the objective (page 6- line 2-21)

Comments of Reviewers:

Literature review:

Overall comment - Review for succinct presentation of ideas and brevity of language. Synthesize and present ideas in summary to the reader removing all and any unncessary information/words (review for brevity)

Comments of Authors: This has been corrected in the revised version. Revised version has been proofread. Correction has been made including typos, comma correction and brevity.

Comments of Reviewers:

Methodology:

The first sentence repeats purpose of study - please remove

Comments of Authors:

Thank you for your comment. This has been corrected in the revised manuscript with track changes (Page 15 - line 1).

Comments of Reviewers:

Subtopic 'Data' can be removed with passage directly under method.

Comments of Authors: Comment is well noted. This has been corrected in the revised manuscript with track changes (Page 14 - line 30)

Comments of Reviewers:

Page 16, conceptual model diagram is repeated and must be removed.

Comments of Authors: Comment is well noted. As per the suggestion, conceptual model sub section has been removed and manuscript adjusted accordingly.

Comments of Reviewers:

Hypothesis should logically be provided under the problem statement or within that section, and not under method.

Comments of Authors: Thank you for your comment. This has been corrected in the revised manuscript with track changes (Page 5 - line 25-32). 

Comments of Reviewers:

Methodology can include measure development information in it as part of it, and results containing all results including demographics from the study - please revise - confusing to reader

Comments of Authors: This has been corrected in the revised version. Now methodological section includes the development information (questioner development and the analytical tool). Results section includes the descriptive statistics, reliability and regression results. 

Comments of Reviewers:

A discussion section is presented before analysis - the use of topics/subtopics are confusing to the reader

Comments of Authors: Thank you for your comment. This has been corrected in the revised manuscript. Result and discussion sections has been separated. 

Comments of Reviewers:

Why is reliability analysis of the tool provided after presenting demographic information of the study population? Please be mindful of the flow of the article

Comments of Authors: This has been corrected in the revised version. demographic information has been now presented after the reliability analysis (Page 19 - line 3-19 and page 20 line 1-4).

Comments of Reviewers:

Clearly demark 'discussion' following results presentation and discuss all matters related to findings under that subtopic

Comments of Authors: Well noted. Subtopic has been introduced under discussion section (Page 28 – line 3; Page 28 – line 17-18; Page 30 – line 4-5)

Comments of Reviewers:

Consluions:

Insert limitations as separate topic¬

Comments of Authors: Thank you for your comment. This has been corrected in the revised manuscript (Page 33 - line 24-31)

Comments of Reviewers: Review overall article for logical flow of presentation and language edit for brevity.

Comments of Authors: This has been corrected in the revised version. Revised version has been proofread. Correction has been made including typos, comma correction and brevity.

---

## [Editor Report · Decision Letter 2]

5 Nov 2021

In Employees’ Favour or Not? - The Impact of Virtual Office Platform on the Work-Life Balances

PONE-D-21-14212R2

Dear Dr. Jayathilaka,

We’re pleased to inform you that your manuscript has been judged scientifically suitable for publication and will be formally accepted for publication once it meets all outstanding technical requirements.

Kind regards,

Kingston Rajiah

Academic Editor

PLOS ONE
---

## [Editor Report · Acceptance letter]

10 Nov 2021

PONE-D-21-14212R2 

In Employees’ Favour or Not? - The Impact of Virtual Office Platform on the Work-Life Balances 

Dear Dr. Jayathilaka:

I'm pleased to inform you that your manuscript has been deemed suitable for publication in PLOS ONE. Congratulations! Your manuscript is now with our production department. 

Kind regards, 

on behalf of

Dr. Kingston Rajiah 

Academic Editor

PLOS ONE